# Mechanism of Antimicrobial Peptides: Antimicrobial, Anti-Inflammatory and Antibiofilm Activities

**DOI:** 10.3390/ijms222111401

**Published:** 2021-10-22

**Authors:** Ying Luo, Yuzhu Song

**Affiliations:** 1College of Life Science and Technology, Kunming University of Science and Technology, Kunming 650500, China; luoying@stu.kust.edu.cn; 2Medical College, Kunming University of Science and Technology, Kunming 650500, China

**Keywords:** antibiofilm, anti-inflammatory, antimicrobial, antimicrobial peptides, mechanism of action

## Abstract

Antimicrobial peptides (AMPs) are regarded as a new generation of antibiotics. Besides antimicrobial activity, AMPs also have antibiofilm, immune-regulatory, and other activities. Exploring the mechanism of action of AMPs may help in the modification and development of AMPs. Many studies were conducted on the mechanism of AMPs. The present review mainly summarizes the research status on the antimicrobial, anti-inflammatory, and antibiofilm properties of AMPs. This study not only describes the mechanism of cell wall action and membrane-targeting action but also includes the transmembrane mechanism of intracellular action and intracellular action targets. It also discusses the dual mechanism of action reported by a large number of investigations. Antibiofilm and anti-inflammatory mechanisms were described based on the formation of biofilms and inflammation. This study aims to provide a comprehensive review of the multiple activities and coordination of AMPs *in vivo*, and to fully understand AMPs to realize their therapeutic prospect.

## 1. Introduction

Since their discovery, antibiotics have been widely used in medicine, food, agriculture, and other fields [1]. Moreover, bacteria have developed antibiotic resistance precisely because of the large-scale use of antibiotics. There are multidrug-resistant bacteria and extensively drug-resistant bacteria [2]. Pathogens causing hospital-acquired infections have become a major challenge to human health with the emergence of drug-resistant bacteria and the expansion of their population [3,4]. The era of “antibiotics” has come to an end, gradually entering the era of “post-antibiotics,” leading to the situation in which drug-resistant bacteria will have no drug exposure [5]. The development of new antibiotics is facing a bottleneck at present [6]. In the last two decades, a few new antibiotics, such as linezolid, baptomycin, and bedaquiline, have been introduced into the market for clinical application [7]. Therefore, new antibiotics or antibiotic substitutes need to be urgently developed [8,9,10,11].

Antimicrobial peptides (AMPs) are a class of small-molecule peptides, usually composed of 12–50 amino acid residues [12,13,14]. At present, more than 3100 natural AMPs have been found [15]. AMPs exist widely in nature and are obtained from bacteria, plants, insects, fish, birds, and other animals [16,17,18]. They are important effectors in the innate immune system and the first line of defense to protect against pathogen infection [19,20,21]. They have no highly conserved sequence, but most of them are short, amphiphilic, and highly cationic molecules [5,22]. They have strong antimicrobial activity against a variety of bacteria, fungi, and viruses [23,24]. They have the advantages of low toxicity to eukaryotic cells, strong thermal stability, high solubility, low molecular weight, and lack of resistance [25]. Therefore, they have potential application in medicine, and several antimicrobial agents are already in clinical trials [15,26].

Natural AMPs are produced by the immune system and participate in regulating the immune system against a wide range of invasive pathogens [27,28]. They are considered to have the potential to replace antibiotics because of their broad-spectrum bactericidal effect. However, in the last decade or so, we have begun to recognize that AMPs, as amphiphilic cations, have biological functions such as immunomodulatory, antimicrobial, anticancer, and wound healing activities [29,30,31]. It is also because these additional functions that AMPs are also known as host defense peptides (HDPs) [30].

Each activity of AMPs has a different mechanism of action [32]. The antimicrobial mechanism of AMPs has been studied, but the findings are not satisfactory because the antimicrobial mechanisms of AMPs are extremely diverse, and recent studies have technical limitations [33,34]. Besides the mechanism of antimicrobial activity, researchers have also focused on anti-inflammatory, anticancer, and other biological activities. To put AMPs into use, we must solve the problem of safety and effectiveness. Therefore, obtaining AMPs with low toxicity and strong activity has always been the pursuit of scientific researchers. However, neither the antimicrobial mechanism of AMPs nor other active mechanisms are fully understood, which limits the development of antimicrobial drugs. In this study, the antimicrobial, antibiofilm, and anti-inflammatory mechanisms of AMPs were reviewed.

## 2. Antimicrobial Mechanism of AMPs

This study mainly introduced two models of pore formation: barrel-stave model and toroidal-pore model [31,33,35]; moreover, the carpet model and detergent-like model exist, with no formation of pores [31,33]. For intracellular effects, the entry into the cell and the target points in the cell are of great research value. AMPs can enter the cytoplasm through direct penetration and endocytosis of the plasma membrane [36,37,38]. The intracellular activities of AMPs are varied, such as binding to nucleic acids, inhibiting the synthesis of nucleic acids and proteins, and affecting the cell cycle [39,40,41]. The intracellular targeting position and the transmembrane method need further exploration. AMPs regulate immune response and play a defensive role in the infected site. In addition, they can also induce the synthesis of pro-inflammatory factors, reduce the inflammatory reaction caused by endotoxins, regulate adaptive immunity, and maintain stability in the body [42,43]. This part is covered in Section 3.2 on anti-inflammatory mechanisms.

### 2.1. Mechanism of Cell Wall Targeting

Peptidoglycan is the main component of the bacterial cell wall, and lipid II is an important part of peptidoglycan synthesis [44]. The peptidoglycan layer is essential for the integrity and survival of bacteria [45]. AMPs (bacitracin and vancomycin) can selectively bind to lipid II, a cell wall synthesis precursor molecule, and inhibit the synthesis of the cell wall (Figure 1) [46,47]. Tanja Schneider et al. performed genetic and biochemical *in vitro* experiments and found that plectasin used lipid II as its cell target and worked by directly binding to lipid II [48]. AMPs not only inhibited the synthesis of the cell wall but also destroyed the formed cell wall structure, as shown in Figure 1. For example, derivative peptide RWRWRW-NH2 destroyed the integrity of the cell wall by affecting respiration and delocalized the cell wall biosynthesis protein MurG [49].

### 2.2. Mechanism of Membrane Targeting

The net charge of cation is an important factor in the early interaction with the negatively charged membrane [50]. Most AMPs are cationic peptides, while G+ and G− surfaces contain teichoic acid and lipopolysaccharides, respectively. Therefore, a net negative charge is generated on the membrane surface. Consequently, the cationic AMPs have initial electrostatic attraction, providing the basis for the next step to destroy the membrane structure or enter the cell to play a role [5]. With the increase in the peptide molecular content, the electrostatic attraction and the penetration of AMPs binding to the cell membrane are strengthened, and then peptide molecules diffuse and pre-assemble freely on the membrane surface [31,33,51]. The transmembrane pore model and the nonmembrane pore model were proposed based on the presence of holes in the membrane structure of AMPs. Each model used different modes of action, but they were related to each other.

#### 2.2.1. Transmembrane Pore Model

The typical transmembrane pore models are the barrel-stave model and toroidal-pore model.

Barrel-stave model: Initially, monomer peptide molecules may undergo conformational changes and be limited to insert into the hydrophobic core of the membrane. When the peptide reaches a certain threshold concentration, oligomers are formed between monomer AMP molecules and further inserted into the hydrophobic core. This process should protect the hydrophilic surface of AMPs from coming in contact with the hydrophobic part of the intima. The hydrophobic region of the peptide chain is toward the membrane and interacts with the membrane lipid, while the hydrophilic region is toward the inner side of the barrel wall to form a channel lumen (Table 1) [52,53,54].

Toroidal-pore model: AMPs were adsorbed to the bilayer at a low concentration. At a high concentration, AMPs vertically inserted into the lipid bilayer induced the membrane phospholipid molecules to bend inward and form pores. The peptide chain is embedded in the hydrophilic and hydrophobic interface and arranged in the inner side of the pore with the lipid bilayer head (Table 1) [52,53,54].

#### 2.2.2. Non-Membrane Pore Model

Carpet model and detergent-like mode: AMPs interacted with negatively charged phospholipids in the outer layer of the membrane and were arranged parallel on the membrane surface to form a “carpet”-like structure. When the AMP concentration exceeds the threshold, peptide molecules automatically rotate and destroy the direction of phospholipid molecules, resulting in enhanced membrane fluidity. The cell membrane is split inward in a way similar to the detergent and the bilayer structure of the membrane finally disintegrates into micelles. This is also known as the detergent-like model (Table 1) [52,53,54].

Besides several common models, many models are used to describe the mechanism of AMPs. For example, the agglutination model is a micellar complex formed by the combination of cationic peptides and outer membrane lipopolysaccharides of G− or cell wall peptidoglycan of G+ (Table 1 and Figure 1). Peptide molecules do not penetrate the cell membrane, induce bacterial cell agglutination, and agglutinate cells, which are then easily phagocytized [55].

**Table 1 ijms-22-11401-t001:** Action model of antimicrobial peptides (AMPs) on the membrane, characteristics of each model, and typical AMPs.

Action Model		Mode of Action	Represents AMPs
Transmembrane pore model	Barrel-stave model	Holes	Alamethicin, pardaxin, and protegrins [56,57,58]
	Toroidal-pore model	Holes	Lacticin Q and melittin [59,60]
Nonmembrane pore model	Carpet model/Detergent-like mode	Splitting	Cecropin P1 and aurein 1.2 [61,62]
	Agglutination model	Devour	Thanatin [55]

### 2.3. Intracellular Targeting Mechanism of Action

#### 2.3.1. Mechanism of Translocation

Many recent studies have shown that AMPs not only are a mode of membrane action but also have intracellular targets. They are also known as nonlytic antimicrobial peptides. The mechanism of intracellular action is still under investigation [63]. When we talk about the intracellular targeting mechanism of AMPs, we should first introduce cell-penetrating peptides (CPPs), which include all peptides with transmembrane transport capacity, whether natural peptides, synthetic peptides, or chimeric peptides [63,64]. AMPs and CPPs are very similar in structure, sequence, and membrane activity [63]. Moreover, some studies have evaluated the antimicrobial activity of CPPs and showed that AMPs could also reach the cytoplasmic target through nonmembrane permeability [40,65,66]. CPPs mainly focus on mammalian cells and are used as cell delivery tools for drugs and biomolecules [67,68]. AMPs are mainly considered as a tool against bacterial infection, which makes similar molecules exist as an independent type. The different effects of CPPs and AMPs may be due to the difference in membrane composition. In addition, the two groups of peptides have large differences in some biological activities, such as different activities for cancer cells. They also have specificity for the selection of action sites [63]. Although they are different, the similarities in their effects on the translocation mechanism are mainly discussed in this manuscript.

##### Energy-Independent Direct Permeation of the Plasma Membrane

1. Formation of instantaneous pores.

Some AMPs (such as proline-rich AMPs) can first gather on the membrane surface and combine with lipids. The transient destruction of the membrane barrier results in the loss of transmembrane potential and the formation of a transient toroidal gap [37]. Consequently, AMPs are transferred to cells and finally act on the target site. In the Shai–Matsuzaki–Huang model, amphiphilic AMPs are initially parallel to the membrane plane and bound on the membrane surface (Figure 1) [69,70,71]. The hydrophobic amino acids of AMPs are inserted into the bilayer membrane, the cationic part of AMPs is combined with the phosphate of the lipid bilayer, and the peptide direction changes from transverse to vertical, forming instantaneous toroidal holes. The model has certain requirements for peptide concentration, which is related to membrane components. If the peptide concentration is too high, the membrane is destroyed or dissolved in a detergent-like manner. When the peptide/lipid ratio is low, AMPs can disturb the membrane structure and reach the cell interior in a transient and nonlethal manner [72]. The disordered toroidal-pore model allows the single peptide to cross the lipid bilayer horizontally under the action of mechanical stress or electric field force (Figure 1). The other peptides maintain the pore structure near the edge of the pores [60,73].

Due to the instability of pores, the permeability of peptides can be increased by increasing the frequency of pore formation and the transfer rate [74]. The pore formation of AMPs is strictly controlled by their size/conformation and specificity [37,75]. Buforin 2 is a typical antimicrobial peptide that enters the cell through transient pore formation. For Buforin 2, the anionic lipid reduces the repulsive force between peptide molecules, leading to the aggregation of peptide molecules so that pores can be maintained at the level that allows peptide molecules to penetrate and flip (Shai–Matsuzaki–Huang model) [74,76]. Indolicidin is a typical nonmembrane targeting peptide, but it has strong antimicrobial activity. It passes through the outer membrane and the inner membrane and binds to DNA via “boat” (disordered toroidal-pore model) or transmembrane (Shai–Matsuzaki–Huang) arrangements [77].

CPPs can form “toroidal pores” or “barrel pores.” The process of the formation of toroidal pores is as follows [78]: CPPs first gather on the inner lobule of the lipid bilayer after entering the cell. They combine with fatty acids in the plasma membrane to form a transient annular pore. Consequently, CPPs cross the plasma membrane and enter the cell. Due to the change in the environment, fatty acids release CPPs and the pores close. For the barrel-pore model, similar to the barrel-stave pore model of AMPs, holes are formed through the molecular structure of the amphoteric peptide, but the gap formed is not stable [79].

2. Direct translocation through membrane instability.

The lipid-phase boundary defect model consists of flat aggregates formed by peptides on the surface of a negatively charged bacterial membrane. The peptide aggregates form rigid and thick lipid regions with the membrane due to the insertion of aromatic residues into the membrane core. The difference in hardness and thickness between the films leads to space defects, leading to the passage of AMPs through the bacterial membrane (Figure 1) [38,80]. Cateslytin, an Arg-rich AMP, aggregates on the membrane surface, leading to membrane boundary defects [38].

For the “carpet” model, some peptides enter the cell because the combination of peptide and membrane accelerates the fluidity of the membrane [78,81]. The direct translocation mechanism through membrane instability requires not only high membrane affinity but also pH gradient and transmembrane potential [82,83]. Although AMPs and CPPs have similar mechanisms of action, they can complement each other in different ways.

##### Energy-Dependent Endocytosis

Endocytosis is a natural and energy-consuming process in all cells. The endocytosis of AMPs is mainly through receptor-mediated transport pathways. Macromolecules enter cells through membrane proteins; studies have shown that the entry of AMPs into cells is also receptor-mediated [36]. Antimicrobial peptide transporter SbmA is a known transporter that mediates the entry of AMPs into cells (Figure 1) [64,84]. Both PR-39 and Bac7 enter cells through SbmA, but the transporter of Drosocin and Apidaecin into the cytoplasm is unknown (Figure 1) [85,86,87,88,89]. The endocytosis of CPPs includes macropinocytosis, clathrin-/caveolin-mediated endocytosis, and clathrin-/caveolin-independent endocytosis [78]. The detailed mechanism of CPP endocytosis is introduced in reference [78].

Besides the mechanism of action of the aforementioned three AMPs, CPPs can be localized through transmembrane localization after internalization, that is “reverse micelle” mechanism. When CPPs come into contact with lipid bilayers, the conformation of peptides changes for incorporation into lipid bilayers, resulting in the invagination of phospholipid bilayers and the formation of reverse micelles. After entering the cell, phospholipid bilayers release peptides into the cytoplasm [78,90]. The binding of basic amino acids to the membrane is the first step of endocytosis [64]. The internalization of CPPs is affected by guanidine, fatty acids, and plasma membrane pH gradient [78]. After the CPPs internalize into the cytoplasm, they are protected by membrane components to ensure that they are not degraded before they reach the target site and exert their biological activity. However, the escape of CPPs from intracellular vesicles is the main limiting factor for their activity [91]. At present, the mechanism of action of AMPs has not been reported. This mechanism can be used as a reference model for similar phenomena in the future.

#### 2.3.2. Intracellular Mechanism of Action

After the AMPs enter the cell membrane and accumulate, they can target intracellular macromolecules and biological processes for further activity [25,39,92,93]. Nonmembrane-targeting AMPs can bind to nucleic acids and proteins; inhibit the process of replication, transcription, and translation; destroy organelles; or affect the enzyme system to disturb the cell cycle and energy metabolism (Table 2 and Figure 1) [37,39,40,42,94,95,96,97,98,99,100,101].

#### 2.3.3. Development and Significance of Intracellular Targeted AMPs

The mechanism of antimicrobial activity of AMPs has been extensively studied, but only a few AMPs are in the advanced clinical antimicrobial stage [30]. Nonlytic AMPs have certain advantages in the clinic. First, AMPs have specific and diverse intracellular targets, which can effectively organize bacterial resistance. Second, AMPs can be used to carry drugs to target cells for targeted therapy [34]. In recent years, thanks to the development of synthetic biology, a large number of modified AMPs have the advantage of becoming clinical drugs. The modified AMPs with low toxicity and strong activity can be carried into cells and are expected to become a new class of potential CPPs.

### 2.4. Dual or Multiple Mechanisms of Action

With the further development of research, many studies have shown that some AMPs have not only a single mode of action but also have multiple mechanisms. A large number of AMPs, whether natural or derived, have been found to have more than membrane-targeting or intracellular effects. They can act on both the membrane and intracellular substances [41,104,105,106,107,108].

The bacterial membrane is destroyed, and the growth of bacteria is inhibited. Hence, whether AMPs still combine with intracellular substances needs to be clarified. Two possible explanations exist for this phenomenon.

The first explanation: The AMPs destroy the bacterial membrane, leading to the leakage of intracellular substances; the leaked intracellular substances absorb the AMPs to protect the undamaged bacterial cells [5,109,110,111]. Recent studies used mathematical models with population and single-cell experiments on LL-37 to prove the formation of a population combined with LL-37 and a growing population that survived because AMPs were isolated by other substances [110,111]. Zhu and others showed that the diffusion coefficient of DNA-binding protein HU and nonendogenous protein Kaede decreased by super-resolution and single-particle tracking method, which finally showed that a close network was formed between high-concentration AMPs, DNA, and 70S polysomes [109]. The experimental results of the PMAP23 peptide showed that the affinity between PMAP23 peptide and dead bacteria was higher than that of living cells. This effect caused dead bacteria to protect living cells by isolating a large number of peptide molecules [5].

Another possible reason is the research on intracellular effects is basically based on independent *in vitro* studies, and the biggest shortcoming of *in vitro* studies is that they cannot fully explain the precise mechanism *in vivo* [25,94,104,112]. *In vitro* experiments cannot timely simulate changes in the real pH value, salt concentration difference, and other components of the immune system *in vivo* [15,113]. The types of AMPs are also diverse in complex organisms. In the internal environment, AMPs may need to cooperate with other AMPs, or coordinate with various factors in the body, or face different internal environments to choose different mechanisms to exert their biological activities [114,115]. The organisms are sophisticated and highly coordinated. Every biological pathway in the organism is restricted by each other. Some studies have shown that the levels of AMPs secreted by frogs are different under the stimulation of different external microorganisms, indicating that the production of AMPs in the body is strictly regulated [116,117]. In addition, many AMPs have variable regions, which make AMPs regulate translocation behavior and target specificity through operating sequences [118,119]. Therefore, AMPs can respond to the external environment and maintain the stability of the internal environment by manipulating variable regions or coordinate with other factors to maintain the internal balance of the body in the face of different external environments.

## 3. Other Mechanisms

### 3.1. Antibiofilm Mechanism

#### 3.1.1. Biofilm Formation Process

Biofilms are composed of complex microbial communities attached to biological or abiotic surfaces and embedded in the matrix produced by proteins and polysaccharides [120,121]. Extracellular polymeric substances (EPSs) contain extracellular polysaccharides, proteins, nucleic acids, and other small cellular molecules [122,123]. The formation and development of biofilm include four stages:

(a) The aggregation or attachment of microorganisms. In this stage, microorganisms continuously gather on the surface of target cells and establish weak interaction with molecules on the surface through van der Waals force, electrostatic force, and hydrophobic interaction. This process is reversible (Figure 2) [124,125].

(b) Microbial adhesion. In this stage, strong and irreversible connections are formed through covalent interaction, and exopolysaccharides are produced. The accumulated microbial colonies are protected by organelles such as extracellular polysaccharides and pili, which enhance the resistance and growth of the community (Figure 2) [124,125].

(c) Development and maturation of a biofilm. In this stage, a stable film structure is formed, and the colonies further adapt to the growth environment under the protection of the biofilm (Figure 2) [124,125].

(d) Biofilm aging. Biofilm depolymerization enables bacteria to scatter on the surface of other cells to enter the next biofilm cycle (Figure 2) [124,125].

#### 3.1.2. Main Mechanism of AMPs against Biofilms

According to the four processes of biofilm formation, the ways to inhibit the formation of biofilms are as follows:

(I) Disruption of the cell signaling system. LL-37 can reduce the attachment of bacterial cells, stimulate twitch movement, and affect the two main quorum-sensing systems of Las and Rhl to influence the formation of biofilms (Table 3 and Figure 2) [126].

(II) Suppression of the alarm system to avoid excessive reactions of bacteria. The exposure of bacteria to amino acid starvation, fatty acid restriction, and other stress environments triggers the upregulation of guanosine tetraphosphate (ppGpp) and pentaphosphate (pppGpp) signal nucleotides and inhibits RNA synthesis [127,128,129]. PpGpp and pppGpp are combined into (p) ppGpp. The bacterial growth and decomposition are suspended, nutrients are transferred to maintain bacterial capacity requirements, and finally, a biofilm is formed [127,128,129]. Peptide 1018 inhibits biofilm formation by blocking the synthesis of (p)ppGpp through enzymes RelA and SpoT (Table 3 and Figure 2) [130]. DJK5 and DJK6 deplete (p)ppGpp from cells to inhibit biofilm formation (Table 3 and Figure 2) [131,132].

(III) Downregulation of the expression of binding protein transport genes responsible for biofilm formation. AMPs can target the severe stress response in Gram-negative and Gram-positive bacteria, or downregulate the genes involved in biofilm formation and binding protein transport [124]. Human β-defensin 3 significantly reduces the expression of icaA and icaD genes (genes responsible for biofilm production) of *Staphylococcus epidermidis* ATCC 35984 and increases the regulation of icaR expression (genes that inhibit the production of biofilms) (Table 3 and Figure 2). The production of biofilm decreases significantly [133,134]. AMP 1037 can reduce group movement, stimulate convulsive movement, and inhibit the expression of many genes related to biofilm formation, thus directly inhibiting biofilm formation (Table 3 and Figure 2) [135]. In addition, some AMPs, such as Nal-P-113 and KW4, can inhibit the formation of biofilms, but the specific mechanism is not clear [94,136].

The way to destroy the formed biofilm is to interfere with the bacterial membrane potential in the biofilm. This can destroy the bacterial membrane to degrade EPSs. Nisin A can affect the membrane potential of methicillin-resistant *S. aureus* biofilm cells, form stable pores, and lead to ATP leakage (Table 3 and Figure 2) [125]. Esculentin-1a destroys the biofilm of *Pseudomonas aeruginosa* through membrane perturbation, that is, it breaks down the extracellular matrix by destroying the cell membrane (Table 3 and Figure 2) [137]. Peptide P1 acts on *Streptococcus mutans* to form irregular biofilms, which can separate cells and extracellular polymeric matrix (Table 3 and Figure 2) [138]. AMPs, such as Temporin-l, CPF-2, and Kassinatuerin-3, were also found to destroy the biofilm. However, the specific mechanism needs further investigation [120,139,140].

In different biofilm stages, the same antimicrobial peptide can exert its biological activity in a corresponding way. For example, peptide G3 can inhibit bacterial adhesion by reducing surface charge, hydrophobicity, membrane integrity, and adhesion-related gene transcription in the initial stage. In the subsequent stage, G3 interacts with extracellular DNA, destroying the 3D structure of mature biofilms and dispersing them (Table 3 and Figure 2) [141].

**Table 3 ijms-22-11401-t003:** AMPs with antibiofilm activity, including the strains and modes of action.

AMPs	Microorganisms	Mechanism of Action	References
LL-37	*Pseudomonas aeruginosa*	Inhibit bacterial adhesion; disruption of cell signaling system	[126]
DJK5 and DJK6	*Pseudomonas aeruginosa*	Suppress the alarm system	[131,132]
1081	A series of G+ and G− (*Pseudomonas aeruginosa*, *Escherichia* *coli*, etc.)	Suppress the alarm system; eradication of mature biofilms	[130]
Human β-defensin 3	*Staphylococcus epidermidis*	Downregulate the expression of binding protein transport genes responsible for biofilm formation	[133,134]
1037	*Pseudomonas aeruginosa*	Downregulate the expression of binding protein transport genes responsible for biofilm formation	[135]
Nisin A	MRSA	Interfere with the bacterial membrane potential in the biofilm	[125]
Esculentin (1–21)	*Pseudomonas aeruginosa*	Interfere with the bacterial membrane potential in the biofilm	[137]
G3	*Streptococcus mutans*	Inhibit bacterial adhesion; degrade EPSs	[141]
P1	*Streptococcus mutans*	Degrade EPSs	[138]

### 3.2. Anti-Inflammatory Mechanism

#### 3.2.1. Mechanism of Inflammation

Inflammation is a defensive reaction caused by harmful stimulation (chemical and physical factors), inflammatory factors (pathogens), or body damage [142,143]. Inflammatory response, including various physiological and pathological processes, is a mechanism to maintain body balance at the cost of a transient decline in tissue function [144].

The study of the anti-inflammatory mechanism of AMPs mainly focuses on the infection by Gram-negative bacteria. Lipopolysaccharide (LPS) is the main component of the outer membrane of G−, which can be used as a protective barrier against the damage of the external environment. LPS consists of three parts: lipid A is composed of glucosamine, phosphate, and fatty acids; o-specific forms of the oligosaccharide polymer chain, and the polysaccharide core connects the first two parts [145]. The chemical structure of LPS can be found in reference [145]. In treating a bacterial infection with conventional antibiotics, the main mechanism is to destroy the structure of the bacterial cell membrane. This leads to bacterial lysis, releases a large amount of LPS, results in the release of pro-inflammatory factors such as TNF-α, triggers local inflammation, and causes diseases such as sepsis [146,147]. Therefore, LPS is considered to be an effective therapeutic target for bacterial infection [148].

An acute inflammatory reaction is caused by pathogen infection and tissue damage in three ways:

(a) Pathogens invade host cells and proliferate in the host body [144,149].

(b) Inflammatory inducers bind to their sensors. Microbial inducers mainly include pathogen-associated molecular patterns (PAMPs) and virulence factors. Virulence factors bind to their specific sensors or PAMPs bind to Toll-like receptors (TLRs) [144,149].

(c) The signaling pathways are activated *in vivo* and inflammatory factors are released, leading to an inflammatory reaction in target tissues affected by inflammatory mediators [144,149].

#### 3.2.2. Anti-Inflammatory Mechanism of AMPs

The anti-inflammatory mechanisms of AMPs may be as follows:

1. Preventing inflammatory inducers from binding to their sensors (Figure 3).

LPS binding to TLR4 is co-catalyzed by lipopolysaccharide-binding protein (LBP) and CD14 [150]. After LPS is released, it first binds to LBP to form an LPS–LBP complex [150,151,152]. LBP is a serum protein that can stimulate and amplify LPS-induced inflammation [153]. The complex targets the CD14 receptor on macrophages. LBP catalyzes multiple rounds of LPS transfer to CD14, and finally, LPS combines with CD14, while the LPS–LBP complex depolymerizes. CD14 transfers LPS to TLR4, activates the TLR pathway, leads to the expression of inflammatory factors, and induces inflammation [150,151,152]. LPS is a pathogen-associated molecular model of Toll-like receptor, and the lipid A component of LPS can activate TLR4 [154]. Lipid A is the conserved structure and active site of LPS [145].

AMPs can exert anti-inflammatory activity in the following three ways:

(a) Neutralizing LPS. Since the polysaccharide core and the phosphate group of LPS are negatively charged, they can be strongly combined with cationic AMPs [155]. Therefore, the alkyl chains of LPS and the nonpolar side chains of AMPs interact through hydrophobic interactions [156]. After binding with LPS, AMPs can neutralize LPS and inhibit the release of inflammatory factors by directly interacting with LPS [157]. Gutsmann et al. showed through biophysical technology that AMPs could transform lipid A from active conformation to inactive a multilamellar structure, so as to neutralize LPS [158]. In addition, Kaconis et al. used a variety of biophysical technologies such as Fourier transform infrared spectroscopy, x-ray diffraction, and freeze-fracture electron microscopy to study the LPS neutralization of a series of synthetic peptides. The results showed that the activity of AMPs in neutralizing LPS was related to the fluidization of the LPS acyl chain, the strong exothermic Coulomb interaction between the two compounds, and the ability to form LPS multilamellar structures [159]. Heinbockel et al. proved using a mouse model that Pep19-2.5 had strong endotoxin neutralization efficiency. Endotoxin is a component of LPS [160]. Similarly, Wilmar Correa et al. studied the binding of Pep19-2.5 to the bacterial cell membrane through thermodynamic analysis and small-angle x-ray scattering. The experimental results showed that Pep19-2.5 combined with the bacterial cell membrane and caused an exothermic reaction [161].

(b) Inhibition of LPS binding to LBP. Most LPS-binding peptides tend to depolymerize LPS oligomers [162]. This leads to the dissociation of LPS oligomers, thereby inhibiting the binding of LPS to LBP. The anti-inflammatory activity of dCATH was studied by fluorescence spectroscopy and flow cytometry. It was found that dCATH induced strong binding with LPS oligomers, led to the depolymerization of LPS oligomers, and inhibited the binding of LPS and LBP [163]. However, some other studies were inconsistent with this statement. According to reference [164], in contrast to the depolymerization of LPS, AMPs induce LPS to cause strong polymerization and form LPS multilamellar structures. Uppu and Haldar studied the binding of QN-PenP peptides to LPS by fluorescence spectroscopy and dynamic light scattering. The results showed that AMPs bound to LPS did not dissociate or promote LPS aggregation and finally neutralized LPS [164]. The content of this step is worthy of further study.

(c) AMPs can combine with LPS competitively and inhibit the transport of LPS. They cause competitive inhibition with CD14, and hence LPS cannot act on TLR4 receptors [165]. The flow cytometry analysis showed that the derived peptide 18-mer LLKKK could effectively bind to CD14 to inhibit the binding of LPS to CD14 (+) cells [165].

Through upstream inhibition, AMPs can effectively prevent the activation of downstream signaling pathways and inhibit the occurrence of inflammation. Some examples are given in Table 4.

In recent studies, a new anti-inflammatory mechanism of AMPs emerged. Lipoproteins/-peptides (LP), a microbial toxin, could induce inflammation by activating TLR2 [166]. Heinbockelet et al. found that Pep19-2.5 interacted with the polar regions of LP and LPS, through primary Coulomb/polar force, interacted with the nonpolar parts of LP and LPS through hydrophobic interaction, and finally neutralized LP and LPS [167]. For AMPs, this is a new mechanism, which needs more research support.

2. Inhibiting and regulating inflammation-related signaling pathways and the expression of transcription factors (Figure 3).

TLR signaling pathway, nuclear factor-kappaB (NF-κB) pathway, and mitogen-activated protein kinase (MAPK) pathway are three important signaling pathways related to the regulation of inflammatory signal transduction. TLRs pathway is recognized and combined by TLRs and PAMPs to activate downstream NF-κB and MAPK pathways [168,169]. TLRs have two signal transduction pathways: one is the MyD88-dependent TLR signal transduction pathway, while the other is the TRIF-dependent pathway. NF-κB pathway regulates the expression of TNF-α, IL-1β, IL-6, and inflammatory chemokines (MIP-lα and MIP-2); also, C-reactive protein and other acute-phase proteins promote inflammation [170,171]. NF-κB is involved in a variety of inflammatory response–related expression and regulation and plays an important role in the inflammatory response. The MAPK pathway includes C-Jun amino-terminal kinase (JNK), p38MAPK, and extracellular-signal-regulated protein kinase (ERK) [172]. After LPS initiates the TLR pathway by identifying TLR4, it induces the phosphorylation of JNK, p38, and ERK to promote inflammation [173,174].

LPS combines with TLR4-activated TLR pathway and activates downstream NF-κB pathway or MAPK pathway. TRIF-dependent signal transduction is related to the endocytosis of activated TLR4 [152]. Therefore, inhibiting the endocytosis of TLR4 is also a mechanism of the anti-inflammatory activity of AMPs. AMPs inhibit TLR4-mediated NF-κB and MAPK pathways, displaying significant anti-inflammatory activities [154]. Examples are given in Table 4.

The anti-inflammatory mechanism after pathogen infection not only protects the host from infection but also induces adaptive immunity, different from the inflammation caused by aseptic tissue injury (Table 4 and Figure 3) [144]. Inflammation is accompanied by the exudation of various inflammatory cells; the formation of inflammatory cell infiltration is also the main component of the inflammatory defense response. AMPs can regulate inflammatory cells and promote them to play an anti-inflammatory role in local inflammation through migration, chemotaxis, and phagocytosis [175,176]. Inducible nitric oxide synthase (iNOS) can use nitric oxide (No) free radicals to cause oxidative stress and assist macrophages in removing invading pathogens [177]. The anti-inflammatory activity of AMPs cannot be accomplished via a single way of action but involve multiple ways.

**Table 4 ijms-22-11401-t004:** AMPs with anti-inflammatory activity and the mechanism of action of each antibacterial peptide.

AMP	Mechanism of Action	References
LL-37	Binds to LPS receptors (CD14 and TLR4) expressed on cells and inhibits TNF-α; neutralizes LPS; suppresses the macrophage pyroptosis that induces the release of pro-inflammatory cytokines; releases neutrophil extracellular traps; stimulates neutrophils to release antimicrobial microvesicles	[178,179,180]
CAP18	Binds to LPS, inhibits the interaction between LPS and LPS-binding protein, and attaches to CD14 molecule, thus inhibiting the expression of LPS-binding CD14 (+) cells to reduce the production of TNF-α by these cells	[165]
dCATH 12-4 and dCATH 12-5	Bind with LPS oligomers leading to the dissociation of LPS aggregates, which prevents LPS from binding to LBP or alternatively to macrophage CD14 receptor	[163]
PA-13	Neutralize LPS; inhibit LPS-mediated TLR activation	[181]
SET-M33 and SET-M33D	Neutralize LPS; reduce the release of TNF-α, IL6, COX-2, and other inflammatory factors	[182,183]
γ-AA	Inhibits LPS-activated TLR4 signal transduction	[184]
OIR3	Inhibits pro-inflammatory factors TNF-α, IL-1β, and IL-6 release	[185]
LB-PG, CA-PG	Inhibit the expression of pro-inflammatory cytokines and chemokines induced by LPS, such as TNF-α, iNOS, MIP-1α, and monocytes	[186]
GW-A2	Inhibits No, iNOS and TNF-α, and IL-6 in LPS-activated macrophages; reduces NF- κB activation increase; inhibits LPS- and ATP-induced NLRP3 inflammasome activation; neutralizes LPS and ATP	[157]
WALK11.3	Inhibits the expression of inflammatory mediators, including No, IL-1β, IL-6, INF- β, and TNF-α; specifically inhibits TLR4 endocytosis	[187]
Ps-K18	Inhibits TLR4-mediated NF- κB pathway, activating innate defense	[154]
Papiliocin	Inhibits expression of the NF- κB pathway	[188]
CLP-19	Inhibits LPS–LBP binding and subsequent MAPK signaling	[189]
CecropinA	Inhibits ERK, JNK, and p38 phosphorylation in the MAPK pathway	[190]
Human beta-defensin (hBD)-3 and hBD-4	Mediate phosphorylation of ERK-1/2 and p38; activate mast cells, degranulate mast cells, and increase vascular permeability, thereby regulating active defense and enhancing anti-inflammatory effects	[176]
IDR-1	Activates FPR1 chemotactic neutrophils to participate in immune regulation	[175]

## 4. Concluding Remarks and Future Directions

New antibacterial drugs need to be urgently found owing to the increasingly serious problem of antibiotic resistance. Since the discovery of AMPs, their antimicrobial activities have been widely studied. For the antimicrobial mechanism of AMPs, the early research mainly focused on the destruction of the bacterial membrane. AMPs also have intracellular activity, and they are used to carry drugs to target cells for treatment. They have been found to have various biological activities. Therefore, they are also used as candidates for anti-inflammatory and immunomodulatory drugs. However, in-depth research has not been conducted on the mechanism of other activities. For example, the anti-inflammatory mechanism of AMPs mainly focuses on the inflammation caused by LPS.

In humans and other higher animals, natural AMPs are released when the body is stimulated or self-injured, and participate in immune regulation to maintain the stability of the internal environment [28]. Therefore, we should fully understand the activities of AMPs and the coordination between AMPs and other factors *in vivo*, so as to avoid the change in activity during drug use. At present, researchers mainly focus on a single mechanism of action *in vitro*. This is not conducive to the exploration of the relationship between multiple mechanisms. Therefore, we need to find a more effective way to deal with the problem at the macro level.

In addition, structure determines function. Size, residue composition, charge, conformation, helicity, hydrophobicity, and amphiphilicity of AMPs all determine the antimicrobial activity [191,192]. However, the scientific problem of “what structure makes AMPs have biological activities such as antibiofilm and immune regulation” still needs to be explored continuously. It is necessary to further study the structure-function relationship of AMPs so as to obtain AMPs with low toxicity, strong activity, and diverse functions.

In conclusion, this review clarified the shortcomings of the current research on the mechanism of AMPs. The findings might contribute to solving the global issue of antibiotic resistance.

## Figures and Tables

**Figure 1 ijms-22-11401-f001:**
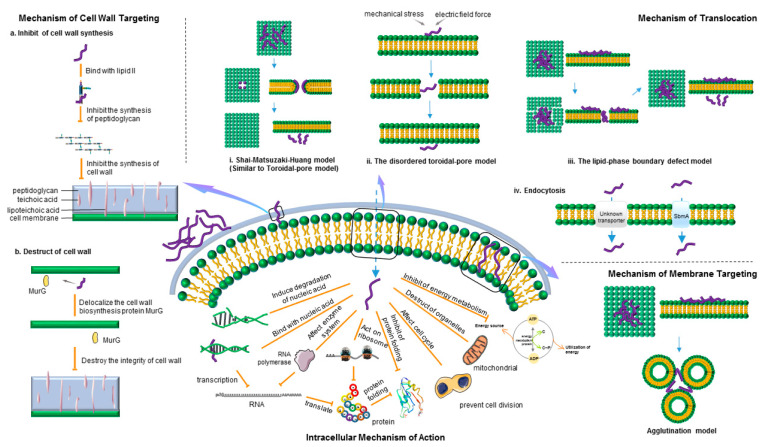
Antimicrobial mechanism of antimicrobial peptides (AMPs). It includes the cell wall–targeting mechanism, membrane-targeting mechanism (only agglutination model is listed), translocation mechanism, and intracellular mechanism of intracellular activity. The blue arrow and yellow line indicate the process, and a short line at the bottom of the yellow line indicates the inhibition (the same below).

**Figure 2 ijms-22-11401-f002:**
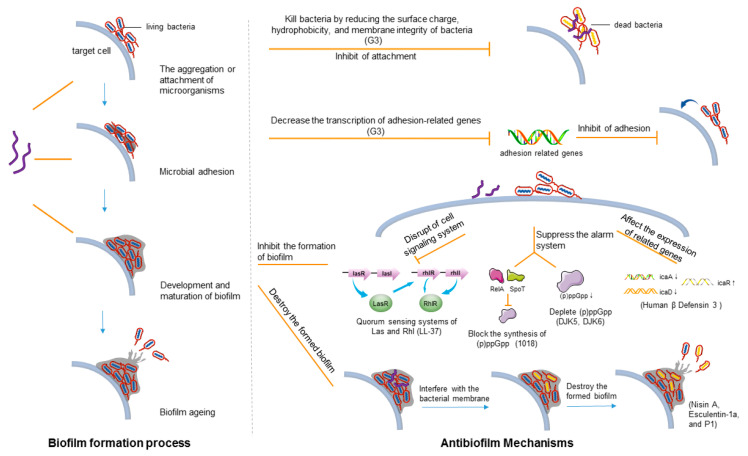
Formation process of biofilms and antibiofilm mechanism of AMPs. The formation of biofilm includes four stages: the aggregation or attachment of microorganisms, microbial adhesion, development, and maturation of biofilm, and aging of biofilm. AMPs can act on these processes to perform biological functions. Upregulation and downregulation of genes are indicated by ↑ and ↓. The AMPs in brackets correspond to the corresponding mechanism types.

**Figure 3 ijms-22-11401-f003:**
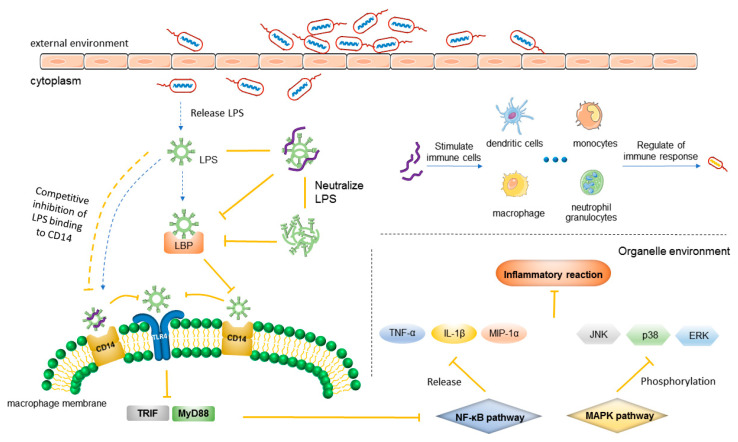
Anti-inflammatory mechanism of AMPs. It mainly refers to the inflammatory reaction caused by LPS. AMPs can inhibit inflammatory pathways (neutralizing LPS, inhibiting the binding of LPS and LBP, competitively inhibiting the binding of LPS and CD14, and inhibiting the release of immune factors), and regulate the immune function of immune cells.

**Table 2 ijms-22-11401-t002:** Summary of the targets, typical AMPs, and specific action modes of AMPs.

Specific Mechanism of Action	AMPs	Action Site	References
Induce degradation of genomic DNA and total RNA	TO17	Nucleic acid	[96]
Bind with nucleic acids and finally inhibit the synthesis of DNA, RNA, and proteins	Buforin-2 and indolicidin	Nucleic acid	[100,101]
Bind with nucleic acids	A series of derived peptides, such as HPA3NT3-A2, MBP-1, IARR-Anal10, and KW4	Nucleic acid	[40,94,102,103]
Bind to RNA polymerase and inhibit the activity of RNA polymerase	Microcin J25 and capistruin	Nucleic acid synthetases	[95]
Act on the termination process of translation. Inhibit protein synthesis by capturing the release factor on the 70S ribosome after hydrolysis of the new polypeptide chain	Apidaecin 1b and Api137	Ribosome	[39]
Transfer of aa-tRNA from EF-Tu to ribosome; a site blocked to inhibit protein synthesis	Bac7, Onc112, pyrrhocoricin, and metalnikowin	Ribosome	[39]
Inhibit the protein synthesis of 70S ribosome and interact with DnaK to inhibit the necessary ATPase activity or protein folding activity	Bac7	Molecular chaperone DnaK	[77]
Inhibit DnaK activity	Abaecin	Molecular chaperone DnaK	[97]
Affect cell cycle, inhibit DNA synthesis, and prevent cell division	Indolicidin	Nucleic acid; cell division	[101]
Affect cell cycle and inhibit cell division	HD5ox	Cell division	[98]
Destruct organelles and inhibit mitochondrial respiration to destroy mitochondria	His-rich AMPs	Mitochondria	[42]
Inhibit the activity of energy metabolism proteins to affect energy metabolism	Magainin 1	Energy metabolism protein	[99]

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
