# Peer review of "Mechanism of Antimicrobial Peptides: Antimicrobial, Anti-Inflammatory and Antibiofilm Activities"

_ijms, 2021, doi:10.3390/ijms222111401_

Round 1
Reviewer 1 Report
Sorry, while the topic is of clear relevance and importance, this manuscript does not feel suitable for publication in a scientific journal. It provides a summary of mechanisms and known facts, which makes it more suitable for a book chapter or similar. Is is nicely written and provides lots of background information, just no scientific advantage required for publication. Also, no recent literature has been discussed, topics lack depth and some statements are simply wrong. Some publications have just been wrongly interpreted.
Author Response
- Response to comment: No recent literature has been discussed.
Response: The previous manuscript focused on the transmembrane mechanism and multiple action mechanism of antimicrobial peptides and ignored the progressive tracking of the latest reports of some contents. We have further improved the membrane action mechanism and anti-inflammatory mechanism of antimicrobial mechanism.
- Response to comment: Topics lack depth.
Response: This paper mainly summarizes the antibacterial (multiple modes of action including intracellular mechanism), anti-inflammatory and antibiofilm mechanisms of antimicrobial peptide. In addition, it also hopes to lead to reader’s concern of the relationship between multiple biological activities of antimicrobial peptides.
- Response to comment: Some statements are simply wrong.
Response: I feel very sorry for my wrong writing. I have corrected the errors according to the opinions pointed out by other reviewers. If there are other errors, I sincerely hope to get your guidance. Thank you very much.
Thank the reviewers again for their comments. I reviewed my article again, which also played a great help in my later study and research.
Reviewer 2 Report
The manuscript by Luo and Song reviews antimicrobial, anti-inflammatory and antibiofilm mechanisms of antimicrobial peptides.
The topic deserves interest but the manuscript and the figures have to be importantly improved before publication.
In general, the reader finds many repetitions, because each new topic is briefly introduced and then explained better in the paragraph. This type of structure brings to too brief and imprecise statements and unnecessary length. I suggest to skip general introductory phrases and go direct to the point with a clear and well-constructed explanation or classification (as an example, page 2 lines 67-73 too general and imprecise).
PAGE 1 “In the last two decades, only three new antibiotics have been introduced into the market for clinical application.” Please name and cite the three antibiotics.
PAGE 3 Description of mechanisms of action have to be made more consistent to what reported in the recent literature. “AMP aggregate and diffuse freely on the surface and then self-assemble on the bacterial membrane” This process can be explained better.
PAGE 2 “AMPs, as amphiphilic cations, have not only an antimicrobial effect but also biological functions such as immunomodulatory, antimicrobial….”AMP have antimicrobial activity, yes.
PAGE 2 “a precursor molecule synthesized by the cell wall”, the cell wall does not synthesize, please rephrase correctly.
Figure 1 is cited before Table 1 and 2 but comes after, that has to be corrected.
Table 1 Function characteristics have to rephrased for the sake of clarity.
PAGE 4 lines 150-154 “The conformations of each kind of peptide are very different; CPPs and AMPs are short cationic peptides, which cannot be distinguished from each other in terms of structure [63,64]. Moreover, some studies have evaluated the antimicrobial activity of CPPs and showed that AMPs could also reach the cytoplasmic target through nonmembrane permeability”. AMP and CPP comparison has to be fully reported, taking into account more recent studies.
“Although some people think that the two are different (WHO?), the transmembrane way of the two is comparable because the ability to reach the inner lobules of the lipid bilayer is crucial to both.” The statement is unclear, better skip it since it is better described at page 5, but please clarify “inner lobule of lipid bilayer”.
“If the concentration of CPPs is very high, they may pass through the cell membrane in a direct ectopic manner [81].” A better explanation is needed.
FIGURE 1. The font of the figure is extremely small and quality of the figure is such that it is impossible to read even by pdf zooming. Besides the figure would be improved if arranged using the same titles used in the text as subheadings. The definition “transmembrane action” is unconventional and, I think, incorrect.
PAGE 7 “First, the AMPs destroy the bacterial membrane, leading to the leakage of intracellular substances; the leaked intracellular substances absorb the AMPs to protect the undamaged bacterial cells. Recent studies have shown that the accumulation of AMPs in cells after cell membrane permeabilization may only result from the combination of intracellular substances and AMPs as other chaperones, which leads to peptide isolation in dead cells. Peptide isolation of dead cells reduces the concentration of effective AMPs so that the remaining bacteria can survive.” The paragraph is cumbersome and contradictory, please improve this session which is new and deserves clearer explanation.
PAGE 7 Lines 282-299. There are a few notable conceptual mistakes:
- “Another possible reason is that the current mechanism of AMPs is by combining artificial lipid membranes with fluorescent dyes.” The statement makes no sense
- “In vitro experiments cannot accurately simulate the local pH, salt concentration difference, small molecules in vivo, and the synergistic relationship between small molecules and other components of the immune system”. PH and salt concentration effects can be much more easily studied in vitro. Which small molecule are meant?
- “AMPs can adjust their sequences according to the environment to achieve their functions. They may choose different mechanisms of action to display their activities or coordinate with other factors…” The statements make no sense, AMP are molecules, they cannot adjust or choose.
PAGE 8. “EPSs, as the protective barriers of bacteria, prevent antibiotics from acting on the cell membrane and reduce the sensitivity of antibiotics, which is the main reason for the antibiotic resistance”. Not true
Biofilm aging (Figure. 2) Give explanation.
Page 8 and 9. Why aren’t all peptides NaI-P113, KW4, Temporin-1, CPF-2 Kassinatuerin not cited altogether. I suggest also to add the branched peptide M33, able to disrupt biofilms (ref Mandarini E et al. Int J Mol Sci. 2020 Nov 5;21(21):8282. doi: 10.3390/ijms21218282; Brunetti J et. Antibiotics. 2020 Nov 24;9(12):840. doi: 10.3390/antibiotics9120840).
PAGE 9 “AMPs can select the corresponding mechanism of action for different stages of bio-film formation” Better rephrase.
FIGURE 2 As for Figure 1, the picture is too low quality and cannot be zoomed in and the font is too small.
PAGE 10 “The anti-inflammatory mechanism of AMPs may involve killing pathogens to prevent their proliferation and spread…” I wouldn’t consider this a proper anti-inflammatory mechanism.
PAGE 10 ***AMPs, are asterisks needed?
PAGE 10 “(a) Neutralizing LPS. AMPs can neutralize LPS and inhibit the release of inflammatory factors by directly interacting with LPS [169]” I suggest also to add the branched peptide M33, able to disrupt biofilms (ref Brunetti J et. Antibiotics. 2020 Nov 24;9(12):840. doi: 10.3390/antibiotics9120840; Brunetti J, J Biol Chem. 2016 Dec 2;291(49):25742-25748. doi: 10.1074/jbc.M116.750257).
FIGURE 3, As for Figure 1 and 2.
English and style, also for the title, have to be extensively adjusted by a professional proof-reader.
Reviewer 3 Report
This is an interesting story about antimicrobial peptides. There are some two weaknesses of the article. In chapter 3.2.2 the LPS binding in an complex interplay between LPS, CD14, LBP TLR4 and other compounds are discussed. The presented mechanisms are much too simple, and essential literature data like those of the group of J. Weiss are missing.
In the same chapter LPS neutralization is discussed. Only the papers of Rosenfeld and Shai are mentioned who write of "LPS depolymerizing" by AMP binding. This is in heavy contrast to many other articles from different groups who came to the conclusion that AMP binding to LPS leads to a strong polymerization. Also, other important details of LPS neutralization comprise exothermic reactions, re-arrangement of the aggregate structure and others. The authors should look at the papers of Gutsmann et al, Kaconis et al, Heinbockel et al, Martinee de Tejada et al, and various others as well as to the peptide 19-2.5 for further details,
Round 2
Reviewer 2 Report
The authors addressed all my concerns in this new version of the manuscript.
Since the review has been extensively modified, bibliography refences have to be carefully re-checked, for example page 5 line 286, number 86 is out of place. Also spaces and all formatting elements have to be checked carefully.
Author Response
Response to comment: Bibliography refences have to be carefully re-checked, for example page 5 line 286, number 86 is out of place. Also spaces and all formatting elements have to be checked carefully.
Response: I am ashamed of my negligence. I have reviewed all the formats and spaces again to make sure they are correct.
Thank you for your valuable advice, which is very important to me.
Reviewer 3 Report
The revised paper again does not fulfill the criteria of a critical review. Thus, the section 3.2 is full of uncritical data and lacking lirerature relevant for an understanding. In particular, the direct binding of AMP to LPS is a fundamental property with high biological consequences. The authors cite literature [174], 2017, for direct interaction to LPS. However other papers were published much earlier and had a stronger impact regarding scientific value . For example, Gutsmann et al, AAC 2010, Kaconis et al Biophys. J. 2011, Heinbockel et al, AAC 2013.
Corresponfding to the uncritical view, the problem of changes of LPS aggregates are discussed. The authors cite papers which come to the conclusion that LPS "depolymerizes" due to AMP binding or "nothing changes" (citations [175, 177, 180] . It is very important to note that changes in fluorescence spectroscopy intensity due to AMP:LPS binding may result from a lot of processes, and not only changes in aggregate structure. It has been shown that the binding of AMP leads to a strong aggregation of LPS (multilamallae) which was shown with more adequate techniques such as X-ray scattering and freeze fracture elecron microscopy.
The authors should consult a paper Heinbockel et al IJMS 2021 in which for a particular AMP the mode of action is presented in detail.
Also, are there no data of AMP binding to Gram-positive toxins such as liporoteins?. Whereas the role of lipoteichoic acids may be doubted, in at least one paper a binding to lipoproteins similar to the role of LPS for Gram-negative bacteria is described (Martinez de Tejada et al, Sci Rep. )
Author Response
1. Response to comment: The section 3.2 is lacking literature relevant for an understanding.
Response: Thank you very much for your suggestions. I'll improve the basic data of this part again. The basic data of Gutsmann et al, AAC 2010, Kaconis et al Biophys. J. 2011, Heinbockel et al, AAC 2013 and others are added. In addition, the composition of LPS and the concepts of LBP and lipid A are supplemented.
2. Response to comment: On the depolymerization of LPS induced by AMPs.
Response: According to the reviewer's suggestion, I will reflect this dispute in this part. See the original text for details.
3. Response to comment:Heinbockel et al, IJMS 2021 on the mechanism of action of AMP binding to Gram-positive toxins such as lipoproteins.
Response: I carefully studied this literature and added the mechanism of Pep19-2.5 binding to Gram-positive toxins (such as lipoproteins) to lines 446-451.
4. Response to comment:Refer to the study of Martinez de Tejada et al, Sci Rep.
Response: This paper is mainly about the combination of polymyxin B nine peptide (PMBN) and efflux pump inhibitor (EPI) to improve antibacterial activity and inhibit bacterial drug resistance. Therefore, it is not cited in this paper.
Thank you for your comments and suggestions. All your suggestions are very important and have important guiding significance for my future scientific research work. According to your comments, I made further modifications. Thank you again for your advice and hope to learn more from you.
Round 3
Reviewer 2 Report
Authors have addressed alla my concerns.
Author Response
Thank you again for all your comments and suggestions. I have learned a lot from it, which will be of great help to my future study and scientific research.
Reviewer 3 Report
The authors have now corrected their erranous citations. It is recommended that in future cited literature should be discussed more critically.
Author Response
Thank you again for your comments and suggestions. Your suggestions and opinions are extremely important to me. In the future study and scientific research, I will continue to cultivate critical thinking and improve my scientific literacy.